# Coping Strategies of Healthcare Professionals with Burnout Syndrome: A Systematic Review

**DOI:** 10.3390/medicina58020327

**Published:** 2022-02-21

**Authors:** Giuseppa Maresca, Francesco Corallo, Giulia Catanese, Caterina Formica, Viviana Lo Buono

**Affiliations:** IRCCS Centro Neurolesi Bonino Pulejo, 98100 Messina, Italy; giusy.maresca@irccsme.it (G.M.); giulia.catanese@ircc.it (G.C.); katia.formica@irccsme.it (C.F.); viviana.lobuono@irccsme.it (V.L.B.)

**Keywords:** burnout, healthcare professional, hospital, coping

## Abstract

*Background and Objectives:* To evaluate the efficacy of coping strategies used to reduce burnout syndrome in healthcare workers teams. *Materials and Methods:* We used PubMed and Web of Science, including scientific articles and other studies for additional citations. Only 7 of 906 publications have the appropriate inclusion criteria and were selected. A PRISMA 2020 flow diagram was used. *Results:* The most common coping strategies that the literature studies showed were efficient, in particular social and emotional support, physical activity, physical self-care, emotional and physical distancing from work. Coping mechanisms associated with less burnout were also physical well-being, clinical variety, setting boundaries, transcendental, passion for one’s work, realistic expectations, remembering patients and organizational activities. Furthermore, it was helpful to listen to the team’s needs and preferences about some types of training. *Conclusion:* We suppose that the appropriate coping strategies employed in the team could be useful also in the prevention of psychological suffering, especially in contexts where working conditions are stressful. Studies about coping strategies to face burnout syndrome in healthcare workers should be increased.

## 1. Introduction

Burnout syndrome has been defined as a chronic response to stress in the workplace [1] characterized by a physical, mental and emotional state of exhaustion [2] that reduces the sense of personal and professional fulfillment [3]. Risk factors could be conflicts and financial problems at work, work overload, communication or organization problems [4]. Some professions are more susceptible than others, and in particular, we focused on studies about healthcare workers that are in daily contact with the seriously ill, such as doctors, nurses and social workers [5]. Howlett et al., [6] highlighted high levels of burnout among the emergency department staff (32.1% suffered from emotional exhaustion), in particular among doctors (46%) which had a high-medium score in burnout scales, and nurses that reported unpleasant contacts with supervisors (they had a high score in burnout scales). Concerns relating to burnout, especially in recent years, have developed a growing interest among mental health scholars. In fact, an important study was recently conducted in Italy that aimed to examine personal resources and psychological symptoms associated with burnout in 933 healthcare workers during the COVID-19 epidemic period. Sociodemographic and occupational data were investigated; depression, anxiety, burnout, and posttraumatic symptoms, as well as psychological well-being, were cross-sectionally assessed using a questionnaire. Results showed a particular incidence of depression (57.9%), anxiety (65.2%), post-traumatic symptoms (55%), and burnout (25.61%) [7]. This syndrome is considered as a multidimensional problem because of a series of symptoms such as depersonalization, anxiety, lack of motivation, mental fatigue, lack of personal and professional achievement, that influence worker and patient’s wellness [5], but it is important to highlight that each person could face problems in different ways. Ding et al., [3] studied the relation between subjective coping and burnout syndrome. Imported findings found significant correlations between emotional exhaustion and emotional and dysfunctional coping, as well as depersonalization and dysfunctional coping Dix, D.M. [8]. The relationship between coping strategies and burnout for caregivers of judged youth. Dysfunctional coping was a significant predictor of burnout [8]. Coping mechanisms are a necessity when dealing with stress and its accompanying stressors. Lazarus and Folkman (1987) classified coping modes as problem-based and emotion-based [9]. Some studies showed the problem-solving approach as the most common coping behavior for health students while the avoidance approach was the least used for coping behaviors in nursing students [10,11,12,13,14,15,16]. Problem-based coping modes are known to be beneficial to students’ learning, clinical performance, and well-being, whereas emotion-based coping modes were found to be detrimental to their health [17,18]. Coping mechanisms and job satisfaction were shown to be associated with the incidence of burnout symptoms in a work setting, according to available literature [19]. There are many differences in job satisfaction between different types of ICUs that are related to patient diagnosis and nursing management [19]. Often the duties and responsibilities of health care workers are not harmonized with the possibilities of the workplace, and training for new tasks is often insufficient [20]. Coping is defined as cognitive and behavioral efforts to manage specific internal and/or external demands that are assessed as taxing or exceeding the person’s resources [21]. A person will be psychologically vulnerable to a particular situation if he or she does not have sufficient coping resources to manage it adequately and places great importance on the threat implicit in the consequences of this inadequate management. Several ways in stress management can be considered, such as cognitive or behavioral coping, cognitive or behavioral avoidance, emotion-focused coping, or substance use [22,23]. From this perspective, burnout can be observed as a progressively developed condition that results from the use of ineffective coping strategies by which professionals attempt to protect themselves from work-related stressful situations [24]. In recent years, the psychological distress and stress of health workers have been studied with interest and attention, with the intent of defining the causes, as well as the causes of stress, have been studied with interest and attention, with the aim of defining the causes, as well as the consequences on a care, organizational and individual level [25]. If, as previously reported, being close to patients is gratifying because it offers the possibility to express different feelings, it is also true that working in hospital wards is extremely demanding and tiring [26]. The quality of the life of healthcare workers is particularly affected by the evident relational asymmetry that is established between the doctor and the patient [27]. Taking care of the suffering of others in increasingly complex organizational contexts, which imply ever-increasing demands for assistance, can induce health workers to raise real defensive barriers against the patient, which leads to extinguishing the flame of passion that animates the doctor and ignites the risk of stress and related pathologies, such as burnout [28]. The aim of this review is to investigate the personal coping strategies of healthcare workers that may have consequent mental health conditions such as burnout syndrome.

We follow the guidelines for the selection of the studies that we identified from each database or register searched (rather than the total number across all databases/registers).

## 2. Materials and Methods

We conducted a narrative review to investigate the efficacy of coping strategies used by healthcare workers to reduce burnout symptoms. Literature studies were performed in accordance to the PRISMA (Preferred Reporting Items for Systematic Reviews and Meta-Analyses) guidelines by searching on PubMed and Web of Science and registered to PROSPERO (ID 312225). We considered the articles from 2008 to 2021. The search combined the following terms: “burnout healthcare professional hospital coping” (all field). There was a total of 844 articles identified via PubMed (Figure 1) and 22 articles from Web of Science. All articles were evaluated by title, abstract, full-text and specificity of the topic (Figure 1). The duplicates were removed, we considered articles that focused on coping strategies of healthcare workers in hospital context.

## 3. Results

We identified 906 studies and seven were selected (Figure 1). All articles conducted research on 1006 healthcare workers with a diagnosis of Burnout Syndrome and investigated the efficacy of coping strategies (Table 1). In particular, the first article highlighted those medical residents who reported low depersonalization, high personal accomplishment, high satisfaction with medicine and high emotional exhaustion after coping strategies, especially social support and entertainment. Koh et al. [29], in their second article, identified coping mechanisms associated with less burnout: physical well-being, clinical variety, setting boundaries, transcendental (meditation and quiet reflection), passion for one’s work, realistic expectations, remembering patients and organizational activities. In their study, Whitebird et al., [30] noted that staff, to manage stress, use physical activity and social support so they could reduce burnout. Mehta et al., [31] evaluated the efficacy of the Relaxation Response Resiliency Program for Palliative Care Clinicians, with positive results (reductions in perceived stress and improvements in perspective-taking). The last article investigated common stressors, coping strategies, and training needs among Palliative Care Clinicians to develop a targeted Resiliency Program. Perez et al., [32] identified three main areas of stressors and coping strategies such as physical self-care, emotional and physical distancing, social and emotional support. Furthermore, the team expressed some needs and preferences: mind–body skills training, cognitive skills, stress education, brief strategies to implement in real-time, enhancing resilience. Two different measures of burnout were identified (Table 2).

Recent studies found a significant correlation between burnout and other variables such as task-focused coping and job satisfaction. Research conducted by Li et al. found that age was positively associated with task-focused coping, job satisfaction, and personal accomplishment, and negatively with secondary traumatic stress, emotional exhaustion, and depersonalization. A very interesting study was conducted with 1027 participants in China, exploring the relationship between coping strategies and job stress [33]. The authors used the Job Performance Scale, the Work Stress Scale, and the Coping Strategies Scale. They determined that the investigated population of healthcare workers employed more positive coping strategies than negative coping strategies and that positive coping strategies mediated the relationship between patient care and job satisfaction, whereas negative strategies moderated the relationship between workload and job performance. Pejuskovic et al. [34] used the Maslach Burnout Inventory and the Ways of Coping scale to assess physicians in Serbia; physicians were also found to be exposed to burnout. These results showed that coping strategies are very important in the development of burnout.

## 4. Discussion

The results of this review, although limited to a few articles in the scientific literature, unlike other studies on this topic, have shown that in work contexts where roles, functions and boundaries are well defined, the mental health of workers is less at risk of developing burnout. It is also important to emphasize that coping strategies, in addition to being influenced by purely personal factors, may also be favored by the work context. Future research should further focus attention on the work climate by promoting individual coping as a resource of the workgroup. Welbourne et al. [35] examined the contribution of occupational attribution style to the use of various coping strategies. Results indicated that the relationship between occupational attribution style and satisfaction was mediated by the use of problem-solving/cognitive restructuring and avoidance strategies to cope with workplace stress. Gracia-Gracia et al. [36] presented results of a correlation between burnout and mindfulness self-compassion in intensive care units. The results of this study showed that the level of burnout is inversely related to their level of self-compassion [36]. A great part of the literature studies focused on the study of burnout and coping strategies in specific healthcare professions (nurses or doctors or social workers).

Healthcare workers often have to face a stressful working environment, especially when they are dealing with the seriously ill and have more responsibilities [4], so this category is particularly at risk of burnout syndrome. However, burnout symptoms seem to be generated principally from systematic and dysfunctional habits and from individual psychological reactions that damage personal wellness, especially emotional exhaustion, depersonalization and reduced personal accomplishment [37]. Instead of implementing his resources to face stressful situations, a worker could react in an “explosive” way (aggressiveness, irritability, attitudes of hostility and resentment), or “implosively” (consequent frustration, chronic anxiety or severe depression). These symptoms could worsen relationships with colleagues and even patients.

In this review, we focused on coping strategies most commonly used to face burnout symptoms in a group context. Although many studies showed personalized multidimensional interventions, the basis of a collaborative climate seems to be a good organization. Sometimes the difficulty could be the lack of habit in teamwork. It might be useful to establish a clear definition of workers’ roles and responsibilities [38], enhancing individual technical competencies. However, only the organization of work is not sufficient. According to Lee et al., [37] a better emotional awareness helped people to feel, understand and express their feelings, thus improving communication. We think that a greater emotional intelligence (the ability to understand the causes of emotions) could help to distinguish subjective and objective problems. Shah et al., [39] highlighted the positive impact of support groups where all staff could meet and discuss the emotional aspects of work, cultivating a sense of “shared understanding”, because a common problem was often the lack or poor communication between colleagues and superiors. Healthcare workers, engaging in self-awareness, regulating emotions, recognizing mistakes and expressing their doubts, could improve empathy and help others. In this way, each worker also could realize that he is not “alone” in the management of stressful situations.

It might be useful to acquire knowledge of appropriate management strategies [39]. For this purpose, in their study, Perez et al., [32] showed that workers spontaneously proposed different solutions, for example, training in mind–body skills, including relaxation exercises such as meditation, breathing, mindfulness group mantra, or yoga; cognitive skills, for example, understanding how to utilize cognitive reframing and strategies to help reduce ruminative thoughts and negative self-talk; a program about stress education that offered information about the physiology of stress and the long-term impact on the body and mind (in this way it was possible to provide a link between physiological signals and distressing thoughts); brief strategies to implement in real-time, such as techniques that helped to organize the day even when the time was limited; learning skills to enhance resilience that would allow them to effectively manage their chronic exposure to stress, improving the care and the relations with patients. This training could be a functional intervention where practice and feedback are essential to produce positive behavioral effects. Furthermore, Perez et al., [32] suggested extending these strategies outside the work environment, and also creating personal spaces for physical self-care, useful for emotional and physical distancing. During challenging periods, some workers expressed the need to briefly disengage from their work to regain composure and preserve psychological equilibrium. For example, they opted to gain physical distancing or simply seek a “time-out” asking for rest periods. In this way, they could spend time on their hobbies, practicing physical activity and taking care of their body, for example, with an adequate nutritional education.

## 5. Conclusions

Considering the findings in all these studies, coping mechanisms have a great influence on the occurrence of burnout, and burnout is highly associated as a significant problem in healthcare institutions. Furthermore, we suggest deepening studies where the healthcare worker’s team is the beneficiary of the appropriate coping strategies, considering that the group could be an important resource to promote collective wellness. Future research should focus further attention on the work climate by making individual coping a resource of the work group.

## Figures and Tables

**Figure 1 medicina-58-00327-f001:**
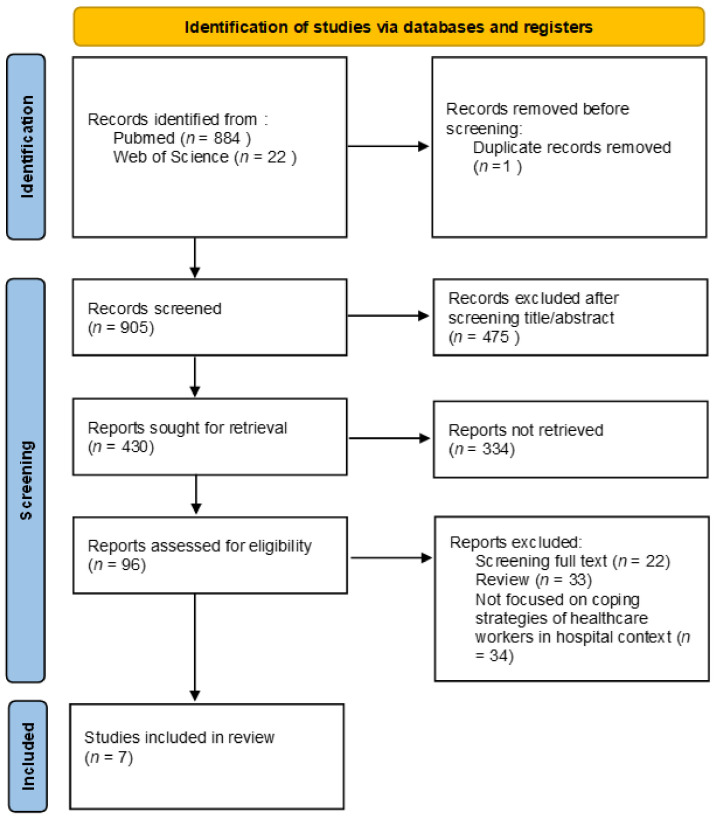
PRISMA 2020 flow diagram of evaluated studies.

**Table 1 medicina-58-00327-t001:** Studies assessing Burnout Syndrome and coping strategies.

References	Aim of the Study	Measures	Socio-Demographic Characteristics	Outcomes
[2]	To show the experiences of stress and burnout and sociodemographic factors associated with dimensions of stress among medical residents	Demographic questions, abbreviated Maslach Inventory, 4 open-ended questions on experiences with stress	136 medical residents (92 man, 44 women)	They responded to the survey, listing an average of 2.2 types of stressors (workload and workplace relationships were the most frequent). They listed an average of 3.1 coping strategies, especially social support and entertainment. Responses indicated low depersonalization, high personal accomplishment, high satisfaction with medicine and high emotional exhaustion
[29]	To estimate the prevalence of burnout and psychological morbidity among palliative care practitioners and its associations with demographic/workplace factors, and with the use of coping mechanisms	Maslach Burnout Inventory –Human Services Survey (MBI-HSS), 12-items General Health Questionnaire (GHQ12)	293 participants (45 Male, 226 women). Age: 20–29 years = 59; 30–39 years= 99; 40–49 years = 67; 50+ years = 44. Profession: 74 Doctors, 156 Nurses, 37 Social worker	The prevalence of burnout among respondents was 91 of 273 (33.3%); psychological morbidity was 77 (28.2%); Home hospice care practitioners (41.5%) were more at risk of developing psychological morbidity. Coping mechanisms associated with less burnout were: physical well-being, clinical variety, setting boundaries, transcendental (meditation and quiet reflection), passion for one’s work, realistic expectations, remembering patients and organizational activities
[30]	To understand how stress affected mental health (in terms of burnout and compassion fatigue) in hospice workers and how they faced these problems	ShortForm12 Health Survey Version 2 (SF-12), Short-form version of the Short-Form 36 Health Survey (SF-36), Generalized Anxiety Disorder (GAD-7) Scale, Patient Health Questionnaire 8 (PHQ8), Professional Quality of Life Assessment R-III Scale (ProQOL-RIII), Short-form version of the Medical Outcomes Social Support Survey (MOS)	547 participants (8% Male, 92% women); Professions: Registered nurses or nurses Practitioners, licensed practical nurses, social workers, home health aides, management/administrative, chaplains/bereavements, volunteer coordinators/others	Hospice staff showed high levels of stress and a small but significant proportion reported moderate to severe symptoms of depression, anxiety, compassion fatigue, and burnout. Staff managed stress through physical activity and social support. These strategies could help decrease staff burnout.
[31]	To evaluate the feasibility of the Relaxation Response Resiliency Program for Palliative Care Clinicians (with the aim of decreasing stress and increasing resiliency)	Perceived Stress Scale, Positive and Negative Affect Schedule, Interpersonal Reactivity Index (IRI), Life Orientation TesteRevised, Satisfaction with Life Scale, General Self-Efficacy Scale	15 participants (3 male, 12 women); Professions: 6 Physicians, 6 Nurses Practitioner Clinical, 2 Social workers, 1 Registered nurse	The intervention was functional. Participants reported reductions in perceived stress and improvements in perspective-taking
[32]	To investigate common stressors, coping strategies, and training needs among Palliative Care Clinicians with the aim of developing a targeted Resiliency Program	Semi-structured interview guide with open-ended questions	15 participants (3 male, 12 women); Professions: 6 Physicians, 6 Nurses Practitioner Clinical, 2 Social workers, 1 Registered nurse	Three main areas of stressors highlighted: challenges related to managing large emotionally demanding caseloads within time constraints; patient factors; personal challenges of delineating emotional and professional boundaries. Coping strategies: physical self-care (i.e., diet, physical activity, sleep, hobbies), emotional and physical distancing, social and emotional support. Training needs and preferences: mind-body skills training, cognitive skills, stress education, brief strategies to implement in real-time, enhancing resilience
[33]	To investigate the effects of coping strategies on the relationship between work stress and job performance for health workers in China	Chinese Nurse Job Stressors Questionnaire	A cross-sectional survey of 852 nurses from four tertiary hospitals in Heilongjiang Province	Positive coping strategies reduce or buffer the negative effects of work stress on job performance and negative coping strategies increased the negative effects.
[34]	To examine correlation between the intensity of Burnout Syndrome and physicians’ personality traits as well as between the level of Burnout Syndrome and stress coping strategies.	Maslach Burnout Inventory, The Temperament and Character Inventory and Manual for the Ways of Coping Questionnaire.	The sample consisted of 160 physicians (70 general practitioners, 50 psychiatrists, 40 surgeons)	Burnout Syndrome affects personal well-being and professional performance.

**Table 2 medicina-58-00327-t002:** Burnout measures.

Burnout Scale	Domains	Items	Scales	Focus
Abbreviated Maslach Burnout Inventory (Maslach C., Jackson S.E. 1981)	Emotional exhaustion; Depersonalization; Personal accomplishment.	9 items	7-point scale	
Maslach Burnout Inventory—Human Services Survey (MBI-HSS) (Maslach C., Jackson S.E. 1981)	Emotional exhaustion; Depersonalization; Personal accomplishment	22 items	7-point scale	To assess an individual’s experience of burnout

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
