# Peer review of "Coping Strategies of Healthcare Professionals with Burnout Syndrome: A Systematic Review"

_medicina, 2022, doi:10.3390/medicina58020327_

Round 1
Reviewer 1 Report
It is an interesting area and the title approach is ok. However, there is little in the text of the approaching strategies. Same goes with the main table of the paper. Figure 1 definitely needs some work as it is hard to read. But I guess the most important thing is that the authors are not very clear in what this review adds to the existing literature. What they are saying in the first phrase of the Discussion section is quite interesting and alright, but most of the review is not strictly following that direction. Maybe some more work in this direction will be desirable. For example how is Table 1 answering to this fact?
Author Response
we have augmented both the introduction section with new references and the discussion section. we have modified table 1. we have tried to clarify the goal of the revision.
Reviewer 2 Report
Dear Authors
Thank you for writing this paper. I must say that although burnout is an important issue to address, I did not understand the rationale to investigate burnout and coping strategies among teams.
You noted in the text: "A negative working environment seems to need a better organization, a clear boundary between the roles of each worker but also dialogue and support between colleagues and superiors. In fact, the opportunity to express needs, preoccupations and preferences could be the basis to plan specific training courses about the knowledge and the
management of stress." - It is unclear and confusing. do you focus on stress or on coping strategies? do you focus on organizational factors or on stress management?
The last sentence in the introduction chapter seems also out of context:
"Pereira et al., [4] also suggested to define a team ethic of care in 51
patient’s assistance and in the relationship with their relatives."
Five papers are not enough to be presented as a systematic review. I would suggest better defining the purpose of the review and broader terms search.
The discussion chapter also seems confusing, mixing organizational/management factors and strategies, with personal strategies, and again - the focus on teams coping strategies is unclear and not well presented.
Author Response
we have accepted your suggestions by removing sentences that are incorrect with respect to what we wanted to discuss. we have expanded the literature. also, the purpose of the review has been clarified.
Reviewer 3 Report
Title - I think that term wellness is not appropriate.
Introduction - I am not sure what is scientific contribution of this literature review. Please better describe what is missing in literature in Italy, neighbor countries, world literature. What was your reason to choose this topic.
Methods - it is not clear how you selected only 5 articles. You should your Prisma flow chart to present flow of your literature review.
Results - I think your results are really strange. Your topic indicate that you rewieving coping strategies but in your 5 articles there is no use of any instrument to measure coping (Ways of Coping scale by Folkman Lazarus). Further on it is not clear how your selected these 5 articles, you have three with quantitative methodology and two with qualitative. You have one article where researchers didn't explore even burnout. I think that you should perform new literature review with clear indication what are you searching for.
Discussion - you start with a coping but in results you do not have coping at all.
Conclusion - not in accordance with results
Author Response
thank you for your valuable suggestions. we have changed the title. we have implemented the literature in the introduction section and have better specified the purpose of the study.
we have fixed the PRISMA diagram and identified other studies.
we have clarified and implemented the results. in the discussion section we have emphasized the importance of coping strategies and clarified the conclusion section.
Round 2
Reviewer 1 Report
I think the authors did a good effort in improving/increasing the quality of the paper. It is definitely much better now. As I said, it is of interest for the readers.
Author Response
thank you for your comments.
Reviewer 2 Report
The authors have addressed and revised their manuscript according to most of the former comments. Yet, the methodology must be improved, the rationale for presenting a review based on 5 papers is not fully explained.
Author Response
we have added studies to increase our results. moreover, the methods section and in the results section we have better described why we have chosen certain articles.
Reviewer 3 Report
Dear authors,
I think that manuscript is improved. However, it is still not clear methodological process of choosing articles for review. There is a lot of literature about burnout and coping strategies, but you have to be strict why you choose this few articles for review.
Please try to be methodological concise.
Author Response
thank you for your suggestions.
we have added studies to increase our results. Moreover, in the methods section and in the results section we have better described why we have chosen certain articles.